Human sickness detection is not dependent on
cultural experience. *Proc. R. Soc. B* **288**:
20210922.

behaviour, cognition

hunter-gatherer, cross-cultural, infectious
disease, facial sickness detection,
disease avoidance, out-group

**Authors for correspondence:**
Artin Arshamian
e-mail: artin.arshamian@ki.se
Asifa Majid
e-mail: asifa.majid@york.ac.uk

†Equal contribution.

Electronic supplementary material is available
online at https://doi.org/10.6084/m9.figshare.
c.5494715.

# Human sickness detection is not dependent on cultural experience

Artin Arshamian[1], Tina Sundelin[1,2], Ewelina Wnuk[3], Carolyn O'Meara[4],
Niclas Burenhult[5,6], Gabriela Garrido Rodriguez[7], Mats Lekander[1,2],
Mats J. Olsson[1], Julie Lasselin[1,2], John Axelsson[1,2,†] and Asifa Majid[8,†]

[1]Department of Clinical Neuroscience, Karolinska Institutet, Stockholm, Sweden
[2]Stress Research Institute, Stockholm University, Stockholm, Sweden
[3]Department of Anthropology, University College London, London, UK
[4]Instituto de Investigaciones Filológicas, National Autonomous University of Mexico in Mexico City, Mexico
[5]Centre for Languages and Literature, and [6]Lund University Humanities Laboratory, Lund University, Sweden
[7]School of Languages and Linguistics, CoEDL, The University of Melbourne, Australia
[8]Department of Psychology, University of York, York, UK

AA, 0000-0003-2282-5903; TS, 0000-0002-7590-0826; EW, 0000-0001-6683-2908;
CO, 0000-0003-2878-8795; MJO, 0000-0001-5592-3759; JL, 0000-0001-8323-0714;
JA, 0000-0003-3932-7310; AM, 0000-0003-0132-216X

Animals across phyla can detect early cues of infection in conspecifics,
thereby reducing the risk of contamination. It is unknown, however, if
humans can detect cues of sickness in people belonging to communities
with whom they have limited or no experience. To test this, we presented
Western faces photographed 2 h after the experimental induction of an
acute immune response to one Western and five non-Western communities,
including small-scale hunter–gatherer and large urban-dwelling communi-
ties. All communities could detect sick individuals. There were group
differences in performance but Western participants, who observed faces
from their own community, were not systematically better than all non-
Western participants. At odds with the common belief that sickness
detection of an out-group member should be biased to err on the side of
caution, the majority of non-Western communities were unbiased. Our
results show that subtle cues of a general immune response are recognized
across cultures and may aid in detecting infectious threats.

## 1. Introduction

Infectious diseases have exerted a heavy selection pressure on most species,
greatly shaping their evolution and their ability to combat infections [1–3].
Most significant diseases throughout human history give rise to salient facial
and bodily cues of illness (e.g. ulcers following plague or rashes from smallpox)
[4]. However, contagious diseases can spread between people long before full
symptom manifestation, and in many cases—such as the one observed in the
pandemic following SARS-CoV2—even before any symptom onset [5].

It has been suggested that the ability to detect cues of infection at an early
stage, and at a safe distance from a sick individual, has been honed so as to
avoid contamination. This ability may be part of a behavioural defence—
often referred to as the behavioural immune system—that enables organisms
to protect themselves against potential pathogens [6–12]. Accordingly, detection
of sick conspecifics is common in the animal kingdom, particularly in group-
living species where infectious diseases can spread quickly [13,14]. In social
insects such as ants, for example, a parasite outbreak initiates the relocation
of the group from the old nest, leaving infected individuals behind [13].
Similarly, chimpanzees shun peers displaying motoric cues of infection [15].

As an ultra-social species [16], humans probably benefit from the ability to detect cues of an infection in others early on, particularly if related to contagious pathogens, and so it has been claimed that the detection of sickness cues from individuals—and, importantly, from strangers—is a fixed phenotype in the metapopulation [17,18]. Although two large online studies have shown that urban dwellers from several cultural regions display similar levels of disgust to photographs of salient bodily skin lesions and computer-generated facial rashes [19,20], little is known about sickness detection per se, and nothing about the universality of such an ability. There is some evidence that people can detect sickness from body odours [18,21], bodily motion [22] and faces [23] when they belong to the same homogeneous group, but cross-cultural data is necessary to uphold claims of universality [24–28]. To address this issue, it is necessary to test sickness detection in (i) small-scale populations that have minimal or no access to technologies in order to mitigate physical or virtual experience with the out-group of interest, and (ii) use natural cues of the initial stages of immune response (i.e. activation of the host defence system).

Today, ease of travel means interaction between groups is rampant, and likewise, contagious diseases spread fast. This can place populations with previously limited exposure to the outside world—and therefore also immune systems naïve to pathogens evolved among visitors—particularly under threat. In the past, this has resulted in the decimation of thriving societies [29]. Data from contemporary societies also demonstrate that infectious disease is the major cause of death in hunter–gatherer and other small-scale populations, in contrast with Western groups who are more likely to die of age-related concerns [30]. Despite this, the current biomedical literature focuses primarily on Western populations, making them a staggering 37 times more likely than non-Westerns to be included in health-related studies. This is extremely problematic as Western populations only account for 11% of the global population [30]. The focus on Western populations is also true regarding research on the detection of sickness. It is therefore critical to ask whether people from diverse backgrounds can detect subtle sickness cues in an infected individual, especially when that infected individual is from an out-group.

To address this, we tested facial sickness detection of Western faces in six different communities across the globe, including people from small-scale indigenous hunter–gatherer societies and large, urban communities. We hypothesized that all non-Western groups would be able to detect a sick face that belonged to a Western out-group and that this would be true even in groups with whom the non-Western group has no or minimal contact. Besides the main aim of the study, we also explored if there were group differences in sickness detection. If sickness detection depends at least partially on experience with particular sickness cues, this would predict that the Western in-group should perform better than all non-Western groups. Furthermore, it has been argued that the strategies for disease-avoidance are a function of information uncertainty, where a higher contamination risk is assigned to out-group over in-group members [10,17]. This 'smoke-detection principle' towards strangers, i.e. a lowered decision criterion for what is perceived as a sickness cue, would reduce the risk for exposure to novel pathogens [10]. Given this, we also assessed whether the same type of heuristic was used across the six groups when they made inferences about the health status of Western faces.

# 2. Material and methods

## (a) Participants

To maximize the social and geographical diversity of test participants, we collected data from three hunter–gatherer and three industrial and post-industrial urban-dwelling communities (figure 1d; for group and culture characteristics, see the electronic supplementary material, text S3). Our first sample of participants came from Stockholm, Sweden ($n = 53$), from the same population as our photograph models. In addition, we tested five non-Western communities. Two came from bustling cities outside of Europe—Ubon, Thailand ($n = 27$), and Mexico City, Mexico ($n = 35$). These are comparable to Stockholm, being large residential units with access to modern technologies, such as television and internet. In addition, we tested three traditionally hunter–gatherer communities, including the forest-dwelling hunter–gatherers Maniq ($n = 18$) and Jahai ($n = 11$) from the equatorial rainforests in the interior of the Malay Peninsula (Thailand and Malaysia), and the hunter–gatherer-fisher Seri ($n = 25$) from the Sonoran coastal desert in northwestern Mexico. These hunter–gatherer communities still live in small groups with limited or extremely limited experience of new faces and infrequent access to television or internet, if at all. Thus, all groups—except the Swedish—made inferences about others' health based on information from an out-group. We collected as many participants as practically possible for the non-Western communities, given that some of these populations are very small. For example, the forest-dwelling Maniq live in groups of 25–35 and have a population of 300, which means that we tested 6% of the total population. No analyses were conducted before data from all participants had been collected (i.e. we did not implement an optional stopping rule, but used a Bayesian approach to address sample size; see Data processing and analysis).

## (b) Acquisition of photos

We used a stimulus set of photographs depicting 13 individuals (Swedish descent) injected under clinical supervision with either *Escherichia coli* (*E. coli*) lipopolysaccharide (LPS; 2.0 ng kg$^{-1}$ bw) or a placebo (saline) (figure 1a; for full procedure see the electronic supplementary material, text S1). LPS activates the innate immune system and induces a distinct—but transient—systemic inflammatory response with symptoms of sickness (e.g. fatigue, headache) [31]. Approximately 2 h after injections, facial photographs were taken. Critically, participants had not reached full symptom manifestation at this point: both body temperature (tympanic) and heart rate reached their peak at approximately 4 h after injection (figure 2). The photographs thus capture the initial phase of infection (figure 1b).

## (c) Procedure

We developed two computer-based rating tasks (figure 1c; electronic supplementary material, text S2) using two photographs each (LPS and placebo) of the 13 participants. In the first task, which was a detection task, participants had to decide whether the person in the photograph was sick or healthy following a yes–no procedure. Each photograph was shown one-by-one and participants were asked in their native language: 'Is this person healthy or sick?' with the response options 'sick' and 'healthy'. This first task is demanding; should participants fail at this they might nevertheless be able to discriminate between a sick versus healthy face when comparing the two directly. To test this, in the second task, the same participants were again shown the facial photos that were used in the first task, but this time with the LPS and placebo photographs of the same individual side-by-side. Participants were asked to indicate which face looked sicker, using a two-alternative forced-choice (2AFC) paradigm.

**3**

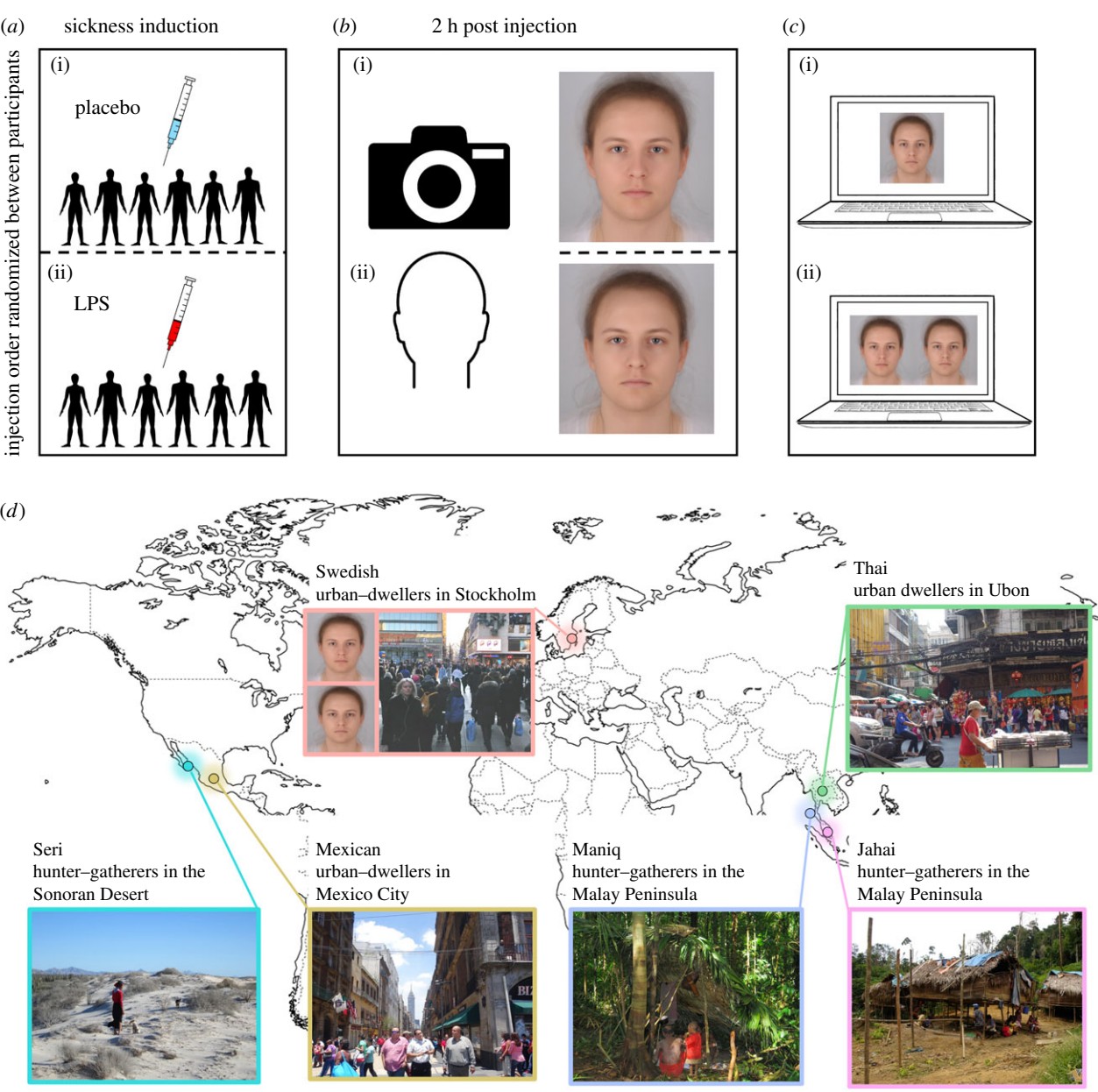

**Figure 1.** Stimulus creation, experimental paradigm and cross-cultural sample. (a) Healthy volunteers were injected with either LPS (E. coli endotoxin) or placebo (saline) on two different occasions in a counter-balanced order. (b) Around 2 h after injection, facial photographs were taken. Participants wore a white t-shirt, no makeup, had their hair away from their face and were told to sit comfortably, look straight into the camera and relax their face. Faces depicted here are average faces for the saline (i) and LPS (ii) conditions. (c) Faces were used in two tasks, a yes–no detection task (i) and a two-alternative forced-choice discrimination task (ii). (d) Six communities were tested with these photographs—three from (post-)industrial, urban settings and three traditional hunter–gatherer communities who live in small-scale groups. The Swedish group constituted the in-group (i.e. making judgements about faces from their own community), while all others were making judgements about out-group faces. (Online version in colour.)

## (d) Data processing and analysis

For the yes–no detection task, we used signal detection theory (SDT), with the unbiased sensitivity measure $d$-prime ($d'$) as a measure of sickness detection [33], where $d'$ is defined as the difference between z-transformed hit (H) and false alarm rate (FA), $d' = z(H) - z(FA)$. Hit and false alarm rates of 1 and 0 were adjusted to $1-1/(2N)$ and $1/(2N)$, respectively, where $N$ is the number of targets/lures (i.e. 13). Values of $d'$ above 0 indicate an ability to detect a signal, 0 indicates performance on the chance level. Negative $d'$ can arise from random distributions or from systematic error but to avoid inflated effects, we did not exclude any participant, even if they had negative $d'$. According to an SDT perspective, participants evaluate targets and distractors on a dimension of signal strength on which the participant set a decision criterion ($c$), or response bias, which indicates the degree of strength that has to be exceeded for an

item to be accepted as a cue of sickness. The criterion is a standard deviation unit measuring the level of preference for answering 'yes' (this person is sick) or 'no' (this person is not sick) and is defined as $c = -1/2 [z(H) + z(FA)]$. Negative values of $c$ indicate a liberal response bias with a tendency to respond 'yes', whereas positive values indicate a conservative response bias with a tendency to respond 'no', and zero indicates a neutral, unbiased response [33]. For the 2AFC discrimination paradigm, proportion correct (Pc) was used, which in this case is the sum of hits and correct rejections divided by the total number of trials.

We used a Bayesian inference approach for estimating $d'$, $c$ and Pc across groups where there is no need to prespecify the sample size [34]. Specifically, for our main contrast of interest (i.e. if people can detect a sick face), we used Bayesian one-sample $t$-tests that the population mean was greater than

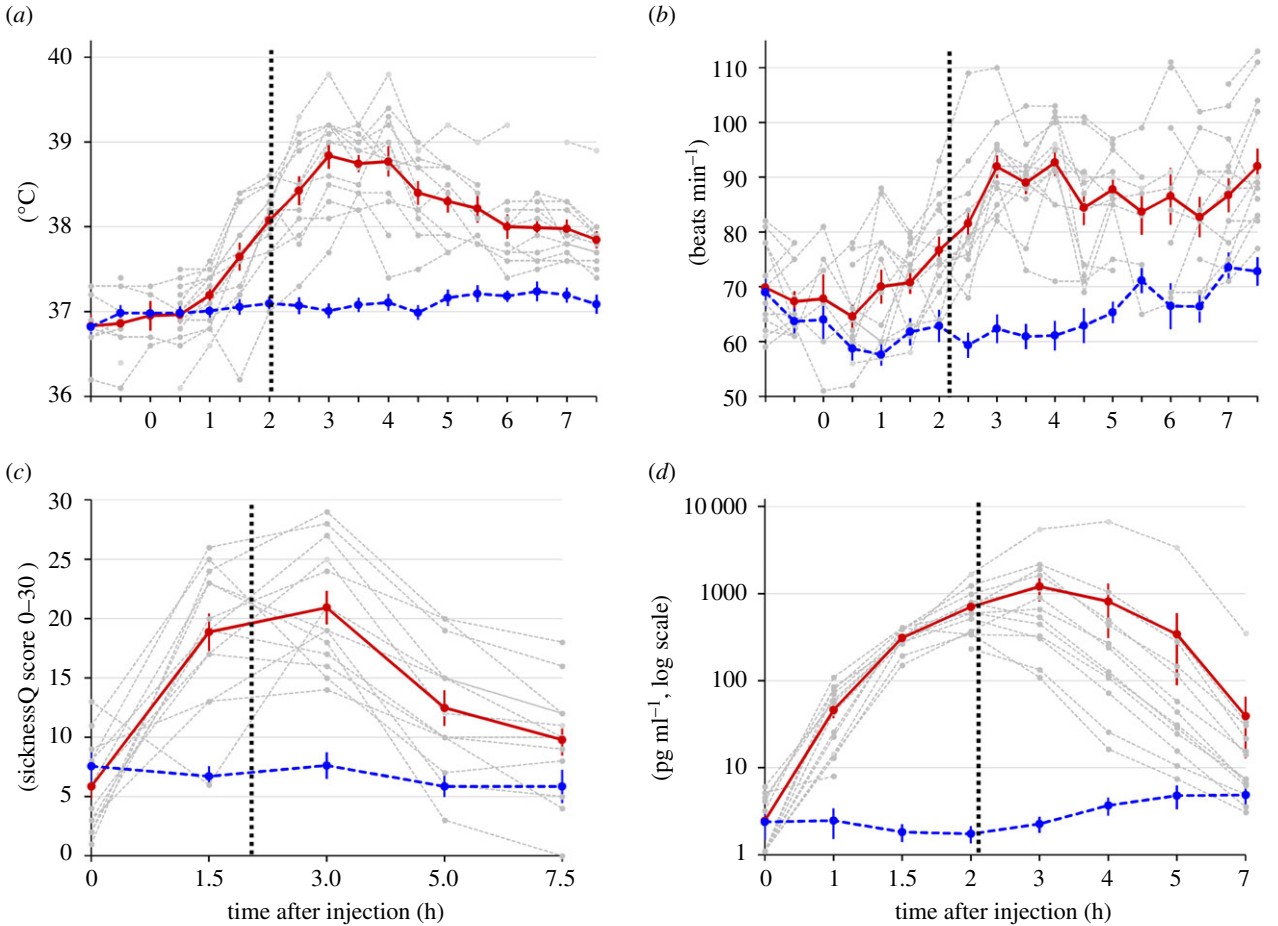

**Figure 2.** Effect of LPS versus placebo (saline) administration on (*a*) body temperature, (*b*) heart rate, (*c*) subjective sickness symptoms, and (*d*) interleukin-6 concentrations. Photos were taken 2 h post injection (dashed horizontal line). Solid red line = LPS administration, mean ± s.e., each individual is shown as a grey dashed line. Dashed blue line = saline, mean ± s.e. The 13 participants participated in both conditions. SicknessQ = sickness questionnaire [32]. (Online version in colour.)

chance (0 for *d′* and 0.5 for Pc) using a Cauchy prior = 0.707 (1/sqrt(2)) [35–37]. In addition to this prior, we also conducted a Bayes factor (BF) robustness check with a wide range of priors and conducted sequential hypothesis testing to estimate how many participants we needed to reach a conclusion about the presence or absence of an effect. This type of analysis typically needs 50–70% smaller samples compared with optimal null hypothesis significance testing [38]. We also controlled for the effect of individual stimuli by calculating the probability that a sick face was categorized as sick compared to the probability that a healthy face was categorized as sick for both the yes–no and the 2AFC task. We did this for both tasks using generalized linear mixed-effects models with a binomial error structure and cross-classified random factors, allowing for random intercepts for rater and facial stimuli (electronic supplementary material, text S4 and tables S11 and S12).

We used Bayesian ANOVAs with the prior *r* scale fixed effects = 0.5; *r* scale random effects = 1 and with *t*-tests using a Cauchy 0, *r* = 1/sqrt(2) prior for the follow-up for individual comparisons [39,40]. The priors used in our analyses place mass in realistic ranges without being overcommitted to any one point. Also, they have been shown to fit a large set of psychological data with moderate effect sizes and convey a minimum degree of information without being uninformative [37,39,41]. The BF depicts an odds ratio, i.e. the probability of the data under one hypothesis relative to another hypothesis. For instance, BF = 4 for H1 indicates that the data are four times more likely under H1 than under H0. The interpretation of the BF followed the standard recommendations

[35,42]. These state that BF between 1 and 3 should be considered to imply no evidence to anecdotal evidence, 3–10 as moderate (with some caveats; a *p* of 0.05 roughly corresponds to a BF of 3 in a null hypothesis significance testing framework), 10–30 as strong, 31–100 as very strong, and BFs from 100 and above as extreme and decisive evidence with no need to conduct further studies on the subject of matter. The analyses and figures were conducted in R [43], Stata 12.1. and in the JASP software package (JASP Team [44]).

## 3. Results

To answer whether people can universally detect sickness from faces, we analysed data from the first (yes–no) task using signal detection with *d*-prime (*d′*) as our measure of sensitivity. This demonstrated that all groups could detect a sick face above chance (Bayesian one-sample *t*-test against chance level (0) for each group separately, with BF supporting the alternative hypothesis): Swedish, BF = $7.07 \times 10^{15}$; Mexican, BF = $1.55 \times 10^{11}$; Thai, BF = $1.23 \times 10^{8}$; Seri, BF = 2484.77, Maniq, BF = 8.36; Jahai, BF = 263.00 (figure 3*a*; electronic supplementary material, tables S1 and S2). BF robustness check demonstrated that the BFs were stable across a wide range of prior distributions, demonstrating the results were robust. Moreover, sequential Bayesian analysis demonstrated that decisive evidence for

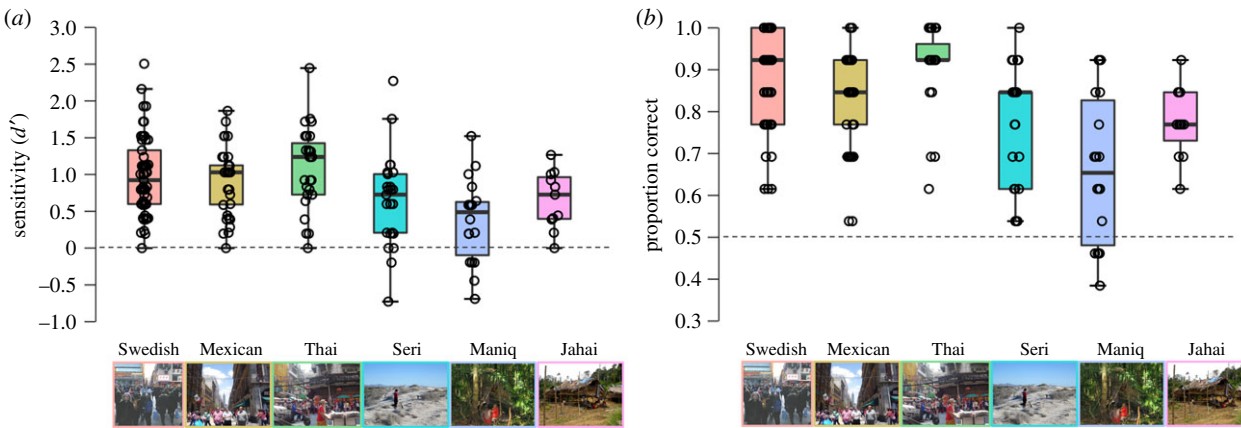

**Figure 3.** Sickness detection and discrimination as a function of culture. (*a*) Boxplots for the yes–no task with $d'$ as the measure of sensitivity, where a value of zero indicates chance performance (horizontal dashed line). (*b*) Boxplots for the two-alternative forced-choice task using proportion correct as the measure of discrimination, with a proportion correct of 0.5 indicating chance level (horizontal dashed line). Boxes indicate the 75th (upper horizontal line), median (middle horizontal line) and 25th (lower horizontal line) percentiles of the distribution. Upper whiskers indicate the maximum value of the variable located within a distance of 1.5 times the interquartile range above the 75th percentile; lower whiskers indicate the corresponding distance to the 25th percentile value. Open circles show individual data points. (Online version in colour.)

sickness detection (i.e. H1 > 100 times more probable than H0; BFs > 100) was reached with 10 participants or less for all groups except the Maniq who reached moderate evidence with 10 participants (electronic supplementary material, figure S1). Next, for the 2AFC task, we used Pc and again all groups could discriminate between a sick and healthy face (Bayesian one-sample $t$-test against chance level (0.5) for each group separately, with BF supporting the alternative hypothesis): Swedish, $BF = 3.91 \times 10^{27}$; Mexican, $BF = 2.20 \times 10^{15}$; Thai, $BF = 2.50 \times 10^{15}$; Seri, $BF = 1.30 \times 10^{7}$; Maniq, $BF = 59.56$; Jahai, $BF = 31309.70$ (figure 3*b*; electronic supplementary material, tables S3 and S4). Importantly, a control analysis showed that our results were not driven by individual stimuli or individual raters (electronic supplementary material, text S4, tables S11 and S12). This indicates that the observed effects would probably generalize to new facial stimuli and speaks to the reliability of the present results. Taken together, the results overwhelmingly favour the hypothesis that across all groups, people can detect and discriminate sick individuals only 2 h after an immune challenge, even if they have little or no experience with that group.

The prediction that the Swedish would be better than all non-Western groups did not hold. Bayesian ANOVA showed decisive evidence that there was a main effect of group, BF = 260.9. However, post hoc tests with corrected posterior odds showed that while there was some variation in absolute sensitivity ($d'$) between groups, the critical prediction (in-group advantage) was not upheld. In fact, the posterior odds showed there was strong evidence for the null hypothesis that the Thai and Mexican did not differ in sickness detection from the Swedish, and inconclusive evidence for the null when comparing the Swedish group to the Jahai and Seri (electronic supplementary material, tables S5 and S6). However, there was very strong evidence for the alternative hypothesis that the Maniq differed from the Swedish group. The 2AFC task showed a similar pattern. Bayesian ANOVA showed decisive evidence for the model with a main effect of group on discrimination between sick and healthy faces, $BF = 5.14 \times 10^{7}$. However, Swedish participants were not uniformly better than other groups. Post hoc tests with corrected posterior odds showed strong evidence for the null

hypothesis that the Thai and Swedish participants did not differ. There was inconclusive evidence that the Swedish participants differed from the Mexican and Jahai (electronic supplementary material, table S7 and S8). However, the Swedish group did better than the hunter–gatherer Seri (strong evidence) and Maniq (conclusive evidence).

Next, to understand the basis of the judgements in more detail, we analysed whether the same type of heuristic was used across the six communities in the detection task. According to the 'smoke-detection principle', the yes–no task in the five non-Western groups should have a response criterion biased toward sick responses over healthy. Swedish participants rating faces from their own community, on the other hand, should display an unbiased, neutral decision criterion. Bayesian one-sample $t$-tests against unbiased criterion (0) for each group separately showed as expected that Swedish participants made unbiased judgements (moderate evidence for the null, $BF_{null} = 6.208$), but so too did the hunter–gatherer Maniq ($BF_{null} = 3.24$), while the Jahai ($BF_{null} = 0.59$) as well as the urban-dwelling Mexicans ($BF_{null} = 1.72$) showed inconclusive evidence for an unbiased response criterion. By contrast, both the Seri (strong evidence, $BF = 38.7$) and Thai (conclusive evidence, $BF = 112.4$) had a biased response criterion with more restrained thresholds for what they accepted as healthy (electronic supplementary material figure S2 and tables S9 and S10). Overall, then, while there were some cross-cultural differences in the ability to detect and discriminate a sick face, there was not universal support for the claim of a lowered decision criterion for what is perceived as sickness in out-group members.

## 4. Discussion

The observation that non-Western groups can detect a sick Western face soon after the initiation of an immune response and do this with similar sensitivity to a Western group is striking, and points to an unexpectedly robust ability in the metapopulation. If our testing had, for example, focused only on Swedish and Maniq, this may have led to the erroneous conclusion that there is a general in-group advantage. However, the fact that Thai and Mexican participants did not differ from Swedish

ones suggests there is no universal in-group advantage for detecting a sick face. In fact, across the two tasks, the Thai were nominally the best.

Taken together, this suggests that sickness detection is based on deciphering infection cues that are shared across people. Although this study does not address what these cues are or whether sickness cues in White-presenting faces may be more discernible than in faces from other communities, recent data from Swedish participants studying the same faces indicate that low-level features like pale skin and droopy eyelids were the most reliable estimates of sickness [23]. Interestingly, post-experimental debriefing with Seri and Mexican participants in the current study suggests droopy eyelids were particularly informative for them as well, indicating that partially similar cues were used. These results are also in line with Tybur et al. [45] who found that most inferences people make about disfigured faces are based on low-level features like coloration. Future studies could manipulate various low-level features and conduct in-depth debriefing to better determine what facial features are particularly diagnostic across communities [46].

The variation in response bias between the non-Western groups is surprising and goes against the 'smoke-detection principle' to always err on the side of caution when evaluating the health of out-group members. However, our findings are consistent with recent theoretical and cross-cultural studies that have challenged the notion of a specific 'smoke-detection principle' for out-group members [12,20,47,48]. Although our data does not address the question of avoidance specifically, the unbiased ratings from most out-groups indirectly support the idea that avoidance is primarily coupled to disease, rather than group status [12,20,47,48]. Moreover, larger social networks have been shown to increase the risk of infectious diseases [49], but the observed variation in response bias between groups did not seem to be a function of group size. In fact, if we use 'residential unit' as a proxy for social network size, the groups with the smallest—Maniq ($N = 25$–$35$) and Jahai ($N = 60$–$70$)—and largest—Mexican from Mexico City ($N = 8.9$ million in the inner city) and Swedish from Stockholm ($N = 950\,000$ in the inner city)—were either clearly unbiased or did not show any evidence of being biased.

Avoidance of unfamiliar and sick conspecifics is common in the animal kingdom, but carer strategies are also widely found [50]. In other words, it is not a given that humans avoid interacting with potentially sick out-group members [51]. In fact, recent mathematical modelling of the evolution of hominin sociality has demonstrated a clear fitness advantage for a carer strategy over sickness avoidance as a method of decreasing disease outbreaks and population crashes [52]. Modelling also shows that although carer strategies emerge at the kin level, once established they spread widely to the broader community [53]. Elaborate caring is something that is evident in traditional hunter–gatherer societies [53]. For example, both the Maniq and Jahai have rich repertoires of rituals targeting disease prevention without an explicit avoidance of sick people [54–56]. Furthermore, both the Maniq and Jahai attribute illness to external factors such as punishment from supernatural forces, unpleasant smells and unusual atmospheric phenomena [54–56]. This kind of reasoning has clear parallels to the history of medicine in the West where infectious diseases were embedded in religious or magical explanations [57]. The notion that sickness is transmitted between humans is thus a relatively modern concept [57].

That being said, as with non-human animals, human disease detection could operate without a disease concept per se. Importantly, even if caring rather than avoidance is more efficient as a strategy, it necessarily relies on recognition of disease cues to preclude infection from one person to another. It should be noted that there was large individual variation in both detection rates and response bias within as well as between groups. Interestingly, Kurves & Wolf [58] also showed higher than expected variation in detection performance for facial sickness with substantially high numbers of expert performers. They also showed large individual variation in response bias for facial sickness detection. Moreover, when they simulated potential social-learning strategies, they found individuals using a 'follow-the-best-member' rule would increase both sensitivity and specificity with increasing group size, but an individual using a 'follow-the-majority' rule would only increase specificity. Unfortunately, we do not know the specific learning strategies participants in our study used. Nevertheless, we believe the mechanisms shaping detection and response bias are multidimensional, including both micro, e.g. inter-personal values [45], as well as macro-level factors, e.g. imitation [59] and culture-specific beliefs [54–56].

The main finding that all groups were able to detect facial sickness is incontrovertible, but some limitations of the current study still need to be addressed. Only White-presenting faces were used as stimuli, which means that any conclusions regarding in-group and out-group sickness detection are confounded by this. Although three different communities (Swedish, Seri and Mexican) reported using similar cues (i.e. droopy eyelids), we do not know if the same features generalize to phenotypically diverse faces. The protocol used to produce the stimuli of sick faces in this study involved injecting Swedish participants with E. coli LPS under medical supervision, but this was not possible to implement in the other global communities tested in this study. Future studies could sample diverse individuals within the West to explore such differences. Moreover, the generalizability of our results is limited by the fact that only 13 individuals contributed to the stimulus set. Although it is reasonable to assume most people's faces would reveal they are sick, it could be the case that some people's facial appearance does not change. Future studies would benefit from a larger and more diverse set of facial stimuli with equally high ecological validity. Importantly, although we had a relatively modest set of facial stimuli, a control analysis demonstrated that our results generalized over them. Specifically, looking at the probability that a sick face was categorized as sick compared to the probability that a healthy face was categorized as sick robustly demonstrated that our results were consistent across stimuli (electronic supplementary material, text S4). Finally, the statistical comparison between groups is naturally affected by the smaller samples from hunter–gatherer communities, which could explain some of the observed group differences. Still, it should be noted that we sampled a rather large contingent of each hunter–gatherer community, which increases the probability that they are more representative of their population as a whole.

To conclude, humans from different parts of the world, whether living in small-scale societies with traditional lifestyles or dwelling in large urban communities with modern technologies, share a common ability to detect cues of sickness based on a general immune response in strangers. The variation found across the five non-Western groups in the heuristics applied to sickness detection indicates that simple

concepts of in-group and out-group alone do not shape decision thresholds for human sickness detection. Instead, being able to detect cues of sickness is likely a robust ability of humans inhabiting diverse cultural contexts.

Ethics. Our research was approved by the Radboud University Ethics Assessment Committee Humanities. The study material and Swedish data collection was approved by the regional ethical review board in Stockholm, Sweden (Registration numbers 2014/1946-31/1, 2015/1415-32) and registered in ClinicalTrials.gov (NCT02529592). All research was carried out in accordance with the provisions of the World Medical Association Declaration of Helsinki.

Data accessibility. Data and code are available at https://osf.io/3mw6x/?view_only=ef7b02faa3c54622b06fc86f3c37bfa2.

Authors' contributions. A.A.: conceptualization, formal analysis, methodology, project administration, visualization, writing—original draft, writing—review and editing; T.S.: formal analysis, investigation, methodology, software, visualization, writing—review and editing; E.W.: investigation, writing—review and editing; C.O.: investigation, writing—review and editing; N.B.: investigation, writing—review and editing; G.G.R.: investigation, writing—review and editing; M.L.: resources, writing—review and editing; M.J.O.: resources, writing—review and editing; J.L.: resources, visualization, writing—review and editing; J.A.: methodology, resources, writing—review and editing; A.M.: funding acquisition, methodology, project administration, supervision, writing—review and editing. All authors gave final approval for publication and agreed to be held accountable for the work performed therein.

Competing interests. We declare we have no competing interests.

Funding. This work was supported by a Netherlands Organization for Scientific Research (NWO) Vici grant (project number 277-70-011) awarded to A.M., who was also supported by an Ammodo KNAW Award and from a grant from the Swedish Foundation for Humanities and Social Sciences Grant (NHS14-1665:1) (to N.B. and A.M.). A.A. was funded by a grant from the Swedish Research Council (2018-01603). M.J.O. was funded by grants from the Swedish Research Council (2012-1125) and (2016-02742), and from the Swedish Foundation for Humanities and Social Sciences grant (P12-1017). M.L. and J.A. were funded by the Stockholm Stress Center (2009-01758).

Acknowledgements. Thanks to Rujiwan Laophairoj (Thai); Karlijn te Paske (Swedish); Hajjah Rogayah A. Razak; the Ministry of Economic Affairs, Putrajaya; and Department of Orang Asli Development (Jahai) for assistance and support with data collection. We also thank Dr Arnaud Tognetti for support in the statistical analysis.

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
