## [Peer Review File · Proceedings of the Royal Society B: Biological Sciences]

Review History

RSPB-2020-2529.R0 (Original submission)

Review form: Reviewer 1

Recommendation

Major revision is needed (please make suggestions in comments)

Scientific importance: Is the manuscript an original and important contribution to its field?

Good

General interest: Is the paper of sufficient general interest?

Good

Quality of the paper: Is the overall quality of the paper suitable?

Marginal

Is the length of the paper justified?

Yes

Should the paper be seen by a specialist statistical reviewer?

Yes

Do you have any concerns about statistical analyses in this paper? If so, please specify them explicitly in your report.

Yes

It is a condition of publication that authors make their supporting data, code and materials available - either as supplementary material or hosted in an external repository. Please rate, if applicable, the supporting data on the following criteria.

Is it accessible?

Yes

Is it clear?

Yes

Is it adequate?

Yes

Do you have any ethical concerns with this paper?

No

Comments to the Author

The current manuscript compares ability to detect ill individuals in western and non-western samples. Illness detection was observed across all samples, presenting evidence against the popular claim that illness detection mechanisms are optimised for detection infectious illnesses in out-group individuals. The study has a lot of strengths. In particular, it's great to see new evidence for illness detection from facial cues that experimentally manipulated illness state and tests perceptions from diverse cultures. Both of these aspects of the study are things it would be good to see more often in the literature. Having said that, the study is let down by two (related issues).

First, the number of individuals in who infection cues were manipulated is very small. This raises questions about the generalizability of the observed effects (i.e., the extent to which the results would generalize to other stimuli) and their reliability. Evidence linking health-related factors to social judgments of faces is equivocal and these mixed results are thought to, at least partly, reflect the fact that findings from studies using small numbers of faces have not replicated in larger studies that tested larger numbers of faces. Ideally, the authors would have created a larger image set prior to data collection.

Second (and relatedly), the analyses do not take into account variation across stimuli. For example, responses on the 2AFC task were aggregated across trials (stimuli) prior to analysis. Although this has been a common approach in much of the face research using this type of paradigm, many researchers have highlighted that this analytical approach (aggregation across stimuli / trials) can cause false positives when the responses are not consistent across stimuli. Mixed effect models in which stimuli is treated as a random, rather than fixed, factor, should be used to address this problem. This analytical approach would go some way to addressing concerns about the small number of stimuli employed.

Review form: Reviewer 2

Recommendation

Major revision is needed (please make suggestions in comments)

Scientific importance: Is the manuscript an original and important contribution to its field?

Good

General interest: Is the paper of sufficient general interest?

Good

Quality of the paper: Is the overall quality of the paper suitable?

Acceptable

Is the length of the paper justified?

Yes

Should the paper be seen by a specialist statistical reviewer?

Yes

Do you have any concerns about statistical analyses in this paper? If so, please specify them explicitly in your report.

No

It is a condition of publication that authors make their supporting data, code and materials available - either as supplementary material or hosted in an external repository. Please rate, if applicable, the supporting data on the following criteria.

Is it accessible?

Yes

Is it clear?

Yes

Is it adequate?

Yes

Do you have any ethical concerns with this paper?

No

Comments to the Author

I have reviewed the manuscript "Human sickness detection is not dependent on cultural experience" with great interest. First of all, I believe that the submitted manuscript is suitable for RSPB and will make a very good contribution to the field. Among the many positive points, it is worth noting the inclusion of small-scale societies, a sound data analytical strategy, and the creation of stimuli with a high degree of ecological validity. On the negative side, the manuscript includes conceptual confusions and other aspects that will require further clarification to be finally accepted for publication.

1. Cues, not signs: The Authors use the terms cues and signs as interchangeably throughout the manuscript. See for example, "signs of illness" (p.4), "detect signs of an infection" (p. 4), "sickness signal" (p. 6), "signal of sickness" (p. 9), "detect signs of sickness" (p. 12), etc for the use of signals and signs, and "detection of sickness cues" (p. 4), "detect subtle sickness cues" (p. 6) or "deciphering infection-signaling cues" (p. 15) for the use of cues.

The theoretical distinction between these two concepts was established by Lorenz (1939) long time ago. In modern ethology, cues and signals are basic concepts that should not be confused (e.g., Smith & Harper, 2003). The present study focuses on cues not on signals.

2. Ability, identification, detection, and judgments: These different concepts appear in the manuscript as if they were referring to the same thing. Please, revise the manuscript to be

consistent. The idea behind the manuscript seems to be built on the assumption of an existing evolved neural mechanism (if we follow Tooby and Cosmides's theorizing) to solve evolutionary problems in the past that is linked to the inference of sickness by observing faces. In sum, we make overall inferences on other's health/sickness by observing their faces.

From an evolutionary point of view, it doesn't matter if our inferences are right or wrong (detection) or if there is a real criterion or not (ability). At the population level, these psychological adaptations will take thousands of years to evolve. However, the adaptiveness of the behaviors (e.g., avoid the "sick" individual) based on the participants' inferences is influenced by the trade offs. These trade offs are influenced by the macro- and micro-context (e.g., living in a small group of people with a very specific social structure), social learning mechanisms (e.g., imitation; see Heyes or Richerson), beliefs (e.g., the Authors rightly mentioned the case of sorcery and sickness), or even group deviations from the norm in in-group/out-group interactions. For example, Kuvers and Wolf (2018) pointed at individual differences and social information use. It is well documented in the field of cultural evolution that we tend to imitate individuals of high prestige and status. Thus, in one society or context, respondents could be making inferences of sickness because their doctor usually provides these cues when they interact (e.g., "your eyes are red"), whereas in other society or context they could be using other cues provided by the sorcerer or medicine man, the leader of the group, etc.

Researchers like Robert Provine or Joshua Tybur have used modified faces manipulating the coloration of the sclera or using disfigured faces. In a recently published article, Tybur et al. (2020) found that most of the judgments ("inferences") were based on low level features (e.g., coloration, skin texture). In sum, the idea of an ability to detect sickness using facial cues is misguided. We make inferences (e.g., Tybur and colleagues refer to "perceived infectability") and we might find more or less agreement (consensus) across and within societies. For example, in Study 2, the Seri and the Maniq provide a clear example of high variability in the inferences being made, whereas the Jahai showed less variability. The Authors mentioned in page 16, "both the Maniq and Jahai attribute illness to external factors such as punishment from supernatural forces, unpleasant smells..." These reflections are very appropriate. It is a pity that the Authors did not conduct debriefing interviews to enquire about which facial cues were used by the participants to classify and discriminate sick faces. In sum, the participants made inferences and there were different levels of agreement on perceived sickness using facial cues.

The last comment is related to the discrimination and classification tasks. Did the same participants complete Studies 1 and 2 using the same set of faces? If so, what was the rationale behind starting with a discrimination tasks first or using the same set of faces. Ideally, a classification task should come first to classify the best models (sick faces) that could be validated in a second stage with a signal detection strategy. If different participants were randomly assigned to only of the studies, please clarify it in the manuscript.

Decision letter (RSPB-2020-2529.R0)

01-Dec-2020

Dear Dr Arshamian:

I am writing to inform you that your manuscript RSPB-2020-2529 entitled "Human sickness detection is not dependent on cultural experience" has, in its current form, been rejected for publication in Proceedings B. This action has been taken on the advice of referees, who have recommended that substantial revisions are necessary. However, the reviewers, Associate Editor and I all appreciate the goals of your manuscript. In particular, I appreciated the breadth of cultures from whom you were able to collect data and the ecological validity of your stimuli.

However, both reviewers note challenges, in particular the issues arising from the small sample of stimuli and some areas of conceptual confusion, that will need some serious consideration. With this in mind we would be happy to consider a resubmission, provided the comments of the referees are fully addressed. However please note that this is not a provisional acceptance.

Sincerely,
 Dr Sarah Brosnan
 Editor, Proceedings B
 mailto: proceedingsb@royalsociety.org

Associate Editor
 Board Member: 1
 Comments to Author:

This paper shows that sickness can be detected from faces in a cross-cultural sample. This has interesting implications for the evolution of faces conveying important information about health, and both reviewers see merit in the questions being asked. However, R1 finds methodological weaknesses in the study, including small sample size of experimental stimuli and variation between stimuli that hasn't been taken into account adequately. R2 has further concerns about some of the conceptual detail (e.g. use cues throughout, not signals). I share these concerns and think that the impact of the paper would be improved if these issues could be addressed.

Referee: 1

Comments to the Author(s)

The current manuscript compares ability to detect ill individuals in western and non-western samples. Illness detection was observed across all samples, presenting evidence against the popular claim that illness detection mechanisms are optimised for detection infectious illnesses in out-group individuals. The study has a lot of strengths. In particular, it's great to see new evidence for illness detection from facial cues that experimentally manipulated illness state and tests perceptions from diverse cultures. Both of these aspects of the study are things it would be good to see more often in the literature. Having said that, the study is let down by two (related issues).

First, the number of individuals in who infection cues were manipulated is very small. This raises questions about the generalizability of the observed effects (i.e., the extent to which the results would generalize to other stimuli) and their reliability. Evidence linking health-related factors to social judgments of faces is equivocal and these mixed results are thought to, at least partly, reflect the fact that findings from studies using small numbers of faces have not replicated in larger studies that tested larger numbers of faces. Ideally, the authors would have created a larger image set prior to data collection.

Second (and relatedly), the analyses do not take into account variation across stimuli. For example, responses on the 2AFC task were aggregated across trials (stimuli) prior to analysis. Although this has been a common approach in much of the face research using this type of paradigm, many researchers have highlighted that this analytical approach (aggregation across stimuli / trials) can cause false positives when the responses are not consistent across stimuli. Mixed effect models in which stimuli is treated as a random, rather than fixed, factor, should be used to address this problem. This analytical approach would go some way to addressing concerns about the small number of stimuli employed.

Referee: 2

Comments to the Author(s)

I have reviewed the manuscript "Human sickness detection is not dependent on cultural experience" with great interest. First of all, I believe that the submitted manuscript is suitable for RSPB and will make a very good contribution to the field. Among the many positive points, it is worth noting the inclusion of small-scale societies, a sound data analytical strategy, and the creation of stimuli with a high degree of ecological validity. On the negative side, the manuscript includes conceptual confusions and other aspects that will require further clarification to be finally accepted for publication.

1. Cues, not signs: The Authors use the terms cues and signs as interchangeably throughout the manuscript. See for example, "signs of illness" (p.4), "detect signs of an infection" (p. 4), "sickness signal" (p. 6), "signal of sickness" (p. 9), "detect signs of sickness" (p. 12), etc for the use of signals and signs, and "detection of sickness cues" (p. 4), "detect subtle sickness cues" (p. 6) or "deciphering infection-signaling cues" (p. 15) for the use of cues.

The theoretical distinction between these two concepts was established by Lorenz (1939) long time ago. In modern ethology, cues and signals are basic concepts that should not be confused (e.g., Smith & Harper, 2003). The present study focuses on cues not on signals.

2. Ability, identification, detection, and judgments: These different concepts appear in the manuscript as if they were referring to the same thing. Please, revise the manuscript to be consistent. The idea behind the manuscript seems to be built on the assumption of an existing evolved neural mechanism (if we follow Tooby and Cosmides's theorizing) to solve evolutionary problems in the past that is linked to the inference of sickness by observing faces. In sum, we make overall inferences on other's health/sickness by observing their faces.

From an evolutionary point of view, it doesn't matter if our inferences are right or wrong (detection) or if there is a real criterion or not (ability). At the population level, these psychological adaptations will take thousands of years to evolve. However, the adaptiveness of the behaviors (e.g., avoid the "sick" individual) based on the participants' inferences is influenced by the trade offs. These trade offs are influenced by the macro- and micro-context (e.g., living in a small group of people with a very specific social structure), social learning mechanisms (e.g., imitation; see Heyes or Richerson), beliefs (e.g., the Authors rightly mentioned the case of sorcery and sickness), or even group deviations from the norm in in-group/out-group interactions. For example, Kuvers and Wolf (2018) pointed at individual differences and social information use. It is well documented in the field of cultural evolution that we tend to imitate individuals of high prestige and status. Thus, in one society or context, respondents could be making inferences of sickness because their doctor usually provides these cues when they interact (e.g., "your eyes are

red”), whereas in other society or context they could be using other cues provided by the sorcerer or medicine man, the leader of the group, etc.

Researchers like Robert Provine or Joshua Tybur have used modified faces manipulating the coloration of the sclera or using disfigured faces. In a recently published article, Tybur et al. (2020) found that most of the judgments (“inferences”) were based on low level features (e.g., coloration, skin texture). In sum, the idea of an ability to detect sickness using facial cues is misguided. We make inferences (e.g., Tybur and colleagues refer to “perceived infectability”) and we might find more or less agreement (consensus) across and within societies. For example, in Study 2, the Seri and the Maniq provide a clear example of high variability in the inferences being made, whereas the Jahai showed less variability. The Authors mentioned in page 16, “both the Maniq and Jahai attribute illness to external factors such as punishment from supernatural forces, unpleasant smells...” These reflections are very appropriate. It is a pity that the Authors did not conduct debriefing interviews to enquire about which facial cues were used by the participants to classify and discriminate sick faces. In sum, the participants made inferences and there were different levels of agreement on perceived sickness using facial cues.

The last comment is related to the discrimination and classification tasks. Did the same participants complete Studies 1 and 2 using the same set of faces? If so, what was the rationale behind starting with a discrimination tasks first or using the same set of faces. Ideally, a classification task should come first to classify the best models (sick faces) that could be validated in a second stage with a signal detection strategy. If different participants were randomly assigned to only of the studies, please clarify it in the manuscript.

Author's Response to Decision Letter for (RSPB-2020-2529.R0)

See Appendix A.

RSPB-2021-0922.R1 (Revision)

Review form: Reviewer 3 (Edward Morrison)

Recommendation

Accept as is

Scientific importance: Is the manuscript an original and important contribution to its field?

Excellent

General interest: Is the paper of sufficient general interest?

Excellent

Quality of the paper: Is the overall quality of the paper suitable?

Excellent

Is the length of the paper justified?

Yes

Should the paper be seen by a specialist statistical reviewer?

No

Do you have any concerns about statistical analyses in this paper? If so, please specify them explicitly in your report.

No

It is a condition of publication that authors make their supporting data, code and materials available - either as supplementary material or hosted in an external repository. Please rate, if applicable, the supporting data on the following criteria.

Is it accessible?

Yes

Is it clear?

Yes

Is it adequate?

Yes

Do you have any ethical concerns with this paper?

No

Comments to the Author

This paper reports a cross-cultural facial perception study showing that people in disparate groups including industrialized and traditional societies can detect infection in stranger's faces.

The paper's main strengths are: the experimental nature of infection; the cross cultural participants; the strong analysis section showing a clear results using signal detection theory and 2 alternative forced choice.

I am very convinced by the results and I believe this is an important contribution to the field.

My one query is about the faces used as stimuli. It would be great to include some (or all) of them in the paper or the supplementary materials to give the reader a sense of what sickness looks like.

This would depend on whether the participants of the infection trial gave permission. But if they did, I would like to see the faces in the paper.

Review form: Reviewer 4

Recommendation

Accept as is

Scientific importance: Is the manuscript an original and important contribution to its field?

Excellent

General interest: Is the paper of sufficient general interest?

Excellent

Quality of the paper: Is the overall quality of the paper suitable?

Excellent

Is the length of the paper justified?

Yes

Should the paper be seen by a specialist statistical reviewer?

No

Do you have any concerns about statistical analyses in this paper? If so, please specify them explicitly in your report.

No

It is a condition of publication that authors make their supporting data, code and materials available - either as supplementary material or hosted in an external repository. Please rate, if applicable, the supporting data on the following criteria.

Is it accessible?

Yes

Is it clear?

Yes

Is it adequate?

Yes

Do you have any ethical concerns with this paper?

No

Comments to the Author

This paper addresses an interesting set of questions (some of which for the first time) on sickness detection within and across cultural groups, such as: How good are people at detecting (ecologically valid) sickness cues? Is performance contingent on one's own cultural background and/or on familiarity with a large number of different groups? And are people biased in their interpretation of cues depending on in-group/out-group affiliation? These questions are investigated in a carefully planned cross-cultural study using facial cues. While the experimental protocol appears well established, the cross-cultural component is novel and expertly designed. I am particularly enthusiastic about the systematic sampling of three hunter-gatherer and three urban populations from several continents, with two of the latter recruited from the same regions as the former. And although sample sizes for the hunter-gatherer groups may appear relatively low, they are actually impressive for such small-scale communities. That the facial pictures are taken from participants experiencing an immune reaction increases ecological validity, which is also an asset of the study reported here.

The findings are partly surprising in that not all support the original hypotheses, but can be seen as all the more convincing given the relative paucity of the visual cues (still pictures of relaxed faces). Had participants been able to observe the full person in real life, with all the additional behavioral cues involved, their performance would most likely have been even better.

Finally, this paper is exceptionally well written, and I very much enjoyed reading it. It has apparently benefited from a previous round of thorough revision (for me, this is the first round with the paper, but I read the responses to reviewers, which address - and fully settle - the only two concerns that I would have had).

So, in conclusion, I consider this paper excellent, both in terms of theoretical novelty and experimental diligence; I regard its findings as interesting and important; and I would love to see it published.

Decision letter (RSPB-2021-0922.R0)

08-Jun-2021

Dear Dr Arshamian

I am pleased to inform you that your Review manuscript RSPB-2021-0922 entitled "Human sickness detection is not dependent on cultural experience" has been accepted for publication in Proceedings B.

The referee(s) do not recommend any further changes. Therefore, please proof-read your manuscript carefully and upload your final files for publication. Because the schedule for publication is very tight, it is a condition of publication that you submit the revised version of your manuscript within 7 days. If you do not think you will be able to meet this date please let me know immediately.

To upload your manuscript, log into <http://mc.manuscriptcentral.com/prsb> and enter your Author Centre, where you will find your manuscript title listed under "Manuscripts with Decisions." Under "Actions," click on "Create a Revision." Your manuscript number has been appended to denote a revision.

You will be unable to make your revisions on the originally submitted version of the manuscript. Instead, upload a new version through your Author Centre.

- 1) A text file of the manuscript (doc, txt, rtf or tex), including the references, tables (including captions) and figure captions. Please remove any tracked changes from the text before submission. PDF files are not an accepted format for the "Main Document".
- 2) A separate electronic file of each figure (tiff, EPS or print-quality PDF preferred). The format should be produced directly from original creation package, or original software format. Please note that PowerPoint files are not accepted.
- 3) Electronic supplementary material: this should be contained in a separate file from the main text and the file name should contain the author's name and journal name, e.g. `authorname_procb_ESM_figures.pdf`

All supplementary materials accompanying an accepted article will be treated as in their final form. They will be published alongside the paper on the journal website and posted on the online figshare repository. Files on figshare will be made available approximately one week before the accompanying article so that the supplementary material can be attributed a unique DOI. Please see: <https://royalsociety.org/journals/authors/author-guidelines/>

4) Data-Sharing and data citation

It is a condition of publication that data supporting your paper are made available. Data should be made available either in the electronic supplementary material or through an appropriate repository. Details of how to access data should be included in your paper. Please see <https://royalsociety.org/journals/ethics-policies/data-sharing-mining/> for more details.

<http://datadryad.org/submit?journalID=RSPB&manu=RSPB-2021-0922> which will take you to your unique entry in the Dryad repository.

Once again, thank you for submitting your manuscript to Proceedings B and I look forward to receiving your final version. If you have any questions at all, please do not hesitate to get in touch.

Sincerely,
Dr Sarah Brosnan
Editor, Proceedings B
<mailto:proceedingsb@royalsociety.org>

Associate Editor
Comments to Author:

I think the authors have done a great job revising this paper and, in my view, this will make a novel and interesting contribution to the literature.

Reviewer(s)' Comments to Author:

Referee: 3

Comments to the Author(s).

This paper reports a cross-cultural facial perception study showing that people in disparate groups including industrialized and traditional societies can detect infection in stranger's faces.

The paper's main strengths are: the experimental nature of infection; the cross cultural participants; the strong analysis section showing a clear results using signal detection theory and 2 alternative forced choice.

I am very convinced by the results and I believe this is an important contribution to the field.

My one query is about the faces used as stimuli. It would be great to include some (or all) of them in the paper or the supplementary materials to give the reader a sense of what sickness looks like.

This would depend on whether the participants of the infection trial gave permission. But if they did, I would like to see the faces in the paper.

Referee: 4

Comments to the Author(s).

This paper addresses an interesting set of questions (some of which for the first time) on sickness detection within and across cultural groups, such as: How good are people at detecting (ecologically valid) sickness cues? Is performance contingent on one's own cultural background and/or on familiarity with a large number of different groups? And are people biased in their interpretation of cues depending on in-group/out-group affiliation? These questions are investigated in a carefully planned cross-cultural study using facial cues. While the experimental protocol appears well established, the cross-cultural component is novel and expertly designed. I am particularly enthusiastic about the systematic sampling of three hunter-gatherer and three urban populations from several continents, with two of the latter recruited from the same regions as the former. And although sample sizes for the hunter-gatherer groups may appear relatively low, they are actually impressive for such small-scale communities. That the facial pictures are taken from participants experiencing an immune reaction increases ecological validity, which is also an asset of the study reported here.

The findings are partly surprising in that not all support the original hypotheses, but can be seen as all the more convincing given the relative paucity of the visual cues (still pictures of relaxed

faces). Had participants been able to observe the full person in real life, with all the additional behavioral cues involved, their performance would most likely have been even better. Finally, this paper is exceptionally well written, and I very much enjoyed reading it. It has apparently benefited from a previous round of thorough revision (for me, this is the first round with the paper, but I read the responses to reviewers, which address – and fully settle – the only two concerns that I would have had). So, in conclusion, I consider this paper excellent, both in terms of theoretical novelty and experimental diligence; I regard its findings as interesting and important; and I would love to see it published.

Decision letter (RSPB-2021-0922.R1)

23-Jun-2021

Dear Dr Arshamian

I am pleased to inform you that your manuscript entitled "Human sickness detection is not dependent on cultural experience" has been accepted for publication in Proceedings B.

Your article has been estimated as being 9 pages long. Our Production Office will be able to confirm the exact length at proof stage.

Data Accessibility section

Open Access

Paper charges

Sincerely,
Editor, Proceedings B
mailto: proceedingsb@royalsociety.org

Appendix A

Dear Dr Brosnan

Thank you for the opportunity to revise our manuscript **RSPB-2020-2529** entitled "**Human sickness detection is not dependent on cultural experience**". We are glad that you, the Associate Editor and the reviewers find our paper to be a significant contribution to this field. We also thank the reviewers and editor for all the great feedback and detailed comments that have considerably improved the paper. We respond to each point below with the editor/reviewer comments in italic font and our response and changes to the manuscript in regular font, but the latter marked in yellow.

On behalf of all the authors,

Artin Arshamian

Associate Editor

Board Member: 1

Comments to Author:

This paper shows that sickness can be detected from faces in a cross-cultural sample. This has interesting implications for the evolution of faces conveying important information about health, and both reviewers see merit in the questions being asked. However, R1 finds methodological weaknesses in the study, including small sample size of experimental stimuli and variation between stimuli that hasn't been taken into account adequately. R2 has further concerns about some of the conceptual detail (e.g. use cues throughout, not signals). I share these concerns and think that the impact of the paper would be improved if these issues could be addressed.

Response: We thank the editor for raising these important points. We have addressed the questions raised by the reviewers and made the appropriate changes. We now control for the variation in facial stimuli, and show that this not change our main findings. We have also removed all references to "signals" and replaced this with cues. We also discuss the theoretical points raised by R2. In addition to the points raised by the reviewers, we now also provide more information of how LPS modulated physiological and psychological factors by adding subjective sickness symptoms and Interleukin-6 levels to Figure 2.

Figure 2. Effect of lipopolysaccharide (LPS) vs. placebo (saline) administration on (a) body temperature, (b) heart rate, (c) subjective sickness symptoms, and (d) interleukin-6 concentrations. Photos were taken 2 hours post injection (dashed horizontal line). Solid red line = LPS administration, mean \pm SE, each individual is shown as a gray dashed line. Dashed blue line = saline, mean \pm SE). The 13 participants participated in both conditions. SicknessQ = Sickness Questionnaire (Andreasson et al., 2018).

Answer to reviewers

Referee: 1

Comments to the Author(s)

The current manuscript compares ability to detect ill individuals in western and non-western samples. Illness detection was observed across all samples, presenting evidence against the popular claim that illness detection mechanisms are optimised for detection infectious illnesses in out-group individuals. The study has a lot of strengths. In particular, it's great to see new evidence for illness detection from facial cues that experimentally manipulated illness state and tests perceptions from diverse cultures. Both of these aspects of the study are things it would be good to see more often in the literature. Having said that, the study is let down by two (related issues).

Response to Reviewer #1:

We thank the reviewer for noting that the study has “a lot of strengths” and pointing to the novelty of the approach we use in this study. We have now addressed the reviewer’s critical points and made the appropriate changes (details below). Our comments are in regular font and the changes to the manuscript are also marked in yellow in the response letter as well as in the manuscript.

Reviewer #1 comments

1. First, the number of individuals in who infection cues were manipulated is very small. This raises questions about the generalizability of the observed effects (i.e., the extent to which the results would generalize to other stimuli) and their reliability. Evidence linking health-related factors to social judgments of faces is equivocal and these mixed results are thought to, at least partly, reflect the fact that findings from studies using small numbers of faces have not replicated in larger studies that tested larger numbers of faces. Ideally, the authors would have created a larger image set prior to data collection.

Response 1:

We acknowledge this limitation and agree that it would have been better with a larger set of photos. In this study, we wanted to use a well-controlled sample of images from the same people when acutely sick as well as healthy. However, this is not a trivial task and we were limited by several factors when making people experimentally sick. First, these studies are very burdensome to subjects and we were only permitted to test a limited number of subjects. Second, the intervention we used to elicit a sickness response is very complex and expensive to carry out. In short, it is very difficult to obtain these kinds of photos, with high ecological validity. It should be noted that other studies have used digitally manipulated faces that easily can be scaled, but a strength of our study is that we used photos of people

with real acute immune response. We have now added a brief discussion of this issue and how it might impact our conclusions. We now write (Pg 19, ln 5-9):

“Moreover, the generalizability of our results is limited by the fact that only 13 individuals contributed to the stimulus set. Although it is reasonable to assume most people’s faces would reveal they are sick, it could be the case that some people’s facial appearance does not change. Future studies would benefit from a larger and more diverse set of facial stimuli with equally high ecological validity.”

2. Second (and relatedly), the analyses do not take into account variation across stimuli. For example, responses on the 2AFC task were aggregated across trials (stimuli) prior to analysis. Although this has been a common approach in much of the face research using this type of paradigm, many researchers have highlighted that this analytical approach (aggregation across stimuli / trials) can cause false positives when the responses are not consistent across stimuli. Mixed effect models in which stimuli is treated as a random, rather than fixed, factor, should be used to address this problem. This analytical approach would go some way to addressing concerns about the small number of stimuli employed.

Response 2

We thank the reviewer for raising this important point. We used a signal detection approach because this allows us to study how active decision-makers make perceptual judgments under conditions of uncertainty. It also enables us to study response bias. However, as the reviewer points out, with these analyses we cannot rule out whether a subset of the faces is driving the effects in our *d*-prime analysis. Therefore, we have re-analyzed the data for the two experiments (Yes-No and 2AFC) using generalized mixed linear models. Using a binomial error structure and cross-classified models with both facial stimuli and rater as random factors, we confirm that all groups could identify sick people above chance. These analyses demonstrate that the ability to detect and discriminate sick faces is not driven by a subset of the facial stimuli. We have added this analysis to the supplementary section and addressed it in the manuscript.

In the methods section we now write (Pg 11-12, ln 19-2):

“We also controlled for the effect of individual stimuli by calculating the probability that a sick face was categorized as sick compared to the probability that a healthy face was

categorized as sick for both the Yes-No and the 2AFC task. We did this for both tasks using Generalized Linear Mixed-Effects Models with a binomial error structure and cross-classified random factors, allowing for random intercepts for rater and facial stimuli (supplemental text S4 and tables S11-12).”

In the discussion we now write (Pg 19, ln 9-13):

“Importantly, although we had a relatively modest set of facial stimuli, a control analysis demonstrated that our results generalized over them. Specifically, looking at the probability that a sick face was categorized as sick compared to the probability that a healthy face was categorized as sick robustly demonstrated that our results were consistent across stimuli (supplemental text S4).”

In the Supplemental results we now write (Pg 7, ln 6-16 and Pg 16):

S4. Control analysis of facial stimuli

The main results from the Yes-No (d') and the 2AFC (P_c) task did not allow us to control for the facial stimuli per se. This is a problem as there is a possibility that people only can detect sickness, and discriminate between health and sickness, in some of the faces. This would effectively question the reliability of our stimuli as both d' and P_c results could be driven by a subset of the facial stimuli. To control for this, we ran separate Generalized Linear Mixed-Effects Models, with binomial error structure for each group, for each task. We used cross-classified models with both rater and facial stimuli as random factors. The results for both tasks (see tables S11-S12) mirrored the results of the signal detection theory-based analyses,

indicating that the outcomes were not driven by particular facial stimuli nor a particular individual within the groups.

Table S11. Effects of cross-classified generalized linear mixed effects models for Yes-No task across cultures

	Odds ratio	SE	95% CI
Swedish	8.45	1.17	6.25 11.59**
Thai	8.31	1.23	5.56 12.73**
Mexican	7.95	1.22	5.46 11.84**
Seri	3.77	1.22	2.57 5.63**
Maniq	1.89	1.22	1.29 2.79*
Jahai	4.47	1.40	2.31 8.64**

Note. Odds ratio of classifying a sick person as sick, with standard error (SE) and 95% confidence interval (CI). *p*-value: * = .001, ** < .001

Table S12. Effects of cross-classified generalized linear mixed effects models for 2AFC task, across cultures

	Probability	Odds ratio	SE	95% CI
Swedish	92%	11.07	1.32	6.69 20.74**

Thai	94%	16.96	1.55	8.06 50.61**
Mexican	84%	5.36	1.26	3.42 9.30**
Seri	79%	3.74	1.24	2.47 6.15**
Maniq	67%	2.07	1.22	1.40 3.05**
Jahai	80%	4.02	1.37	2.17 7.44**

Note. Probability and odds ratio of identifying the sick photograph as sick, with standard error (SE) and 95% confidence interval (CI). p -value: * < .001

Referee: 2

Comments to the Author(s)

I have reviewed the manuscript "Human sickness detection is not dependent on cultural experience" with great interest. First of all, I believe that the submitted manuscript is suitable for RSPB and will make a very good contribution to the field. Among the many positive points, it is worth noting the inclusion of small-scale societies, a sound data analytical strategy, and the creation of stimuli with a high degree of ecological validity. On the negative side, the manuscript includes conceptual confusions and other aspects that will require further clarification to be finally accepted for publication.

Response to Reviewer #2:

We are delighted the reviewer thinks the paper would make a very good contribution to the field and is suitable for publication in RSPB because of its use of data from small-scale societies and methodological rigour. We are grateful to the reviewer for pointing out the infelicities in our writing that created conceptual confusions and have addressed the reviewer's questions and made the appropriate changes. Our comments are in regular point and the changes are marked in yellow in the response letter as well as in the manuscript.

Reviewer #2 comments

1. Cues, not signs: The Authors use the terms cues and signs as interchangeably throughout the manuscript. See for example, "signs of illness" (p.4), "detect signs of an infection" (p. 4),

“sickness signal” (p. 6), “signal of sickness” (p. 9), “detect signs of sickness” (p. 12), etc for the use of signals and signs, and “detection of sickness cues” (p. 4), “detect subtle sickness cues” (p. 6) or “deciphering infection-signaling cues” (p. 15) for the use of cues.

The theoretical distinction between these two concepts was established by Lorenz (1939) long time ago. In modern ethology, cues and signals are basic concepts that should not be confused (e.g., Smith & Harper, 2003). The present study focuses on cues not on signals.

Response 1:

The reviewer is absolutely right that these two terms should not be confused with each other. We have now replaced every instance in the manuscript where the words “signals” and “signs” appear with “cues”.

2. Ability, identification, detection, and judgments: These different concepts appear in the manuscript as if they were referring to the same thing. Please, revise the manuscript to be consistent.

Response 2:

Again, the reviewer is absolutely right. Throughout the manuscript we now use detect/detection instead of identification and judgments.

3. The idea behind the manuscript seems to be built on the assumption of an existing evolved neural mechanism (if we follow Tooby and Cosmides’s theorizing) to solve evolutionary problems in the past that is linked to the inference of sickness by observing faces. In sum, we make overall inferences on other’s health/ sickness by observing their faces.

From an evolutionary point of view, it doesn’t matter if our inferences are right or wrong (detection) or if there is a real criterion or not (ability). At the population level, these psychological adaptations will take thousands of years to evolve. However, the adaptiveness of the behaviors (e.g., avoid the “sick” individual) based on the participants’ inferences is influenced by the trade offs. These trade offs are influenced by the macro- and micro-context (e.g., living in a small group of people with a very specific social structure), social learning mechanisms (e.g., imitation; see Heyes or Richerson), beliefs (e.g., the Authors rightly mentioned the case of sorcery and sickness), or even group deviations from the norm in in-group/out-group interactions. For example, Kuipers and Wolf (2018) pointed at individual differences and social information use. It is well documented in the field of cultural evolution that we tend to imitate individuals of high prestige and status. Thus, in one society or context, respondents could be making inferences of sickness because their doctor usually provides these cues when they interact (e.g., “your eyes are red”), whereas in other society or

context they could be using other cues provided by the sorcerer or medicine man, the leader of the group, etc.

Response 3:

We agree these are all very important points, and we have tried to incorporate them in the manuscript. We now state that people make “inferences”, and refer to the role of imitation and individual differences that could contribute to how sickness detection is shaped. The reviewer correctly remarks that people in this study make “inferences” about perceived sickness or infectability. In an evolutionary context the only interesting thing for survival would have been the “adaptiveness” of their behavior, that is, if sickness detection to some extent is inherited then a phenotype that helps the organism to stay healthy could also have contributed to increase that organism’s fitness (e.g., increased the number reproduction cycles). We did not, however, ask if participants would have avoided or approached the depicted individuals if they would have met them in real life. Even if we had done that, we do not know if that specific phenotype would have increased fitness. The only thing we can claim based on the current data is that people could reliably detect (Yes-No task) and discriminate (“2AFC) cues of objective measures of acute immune response (i.e., their inferences about the faces were in most cases objectively correct). However, it should be noted that in an evolutionary context it does of course matter if the inferences are based on something real and substantial. If all unknown faces were deemed to be sick then that would be relatively costly (independent of the phenotype: avoid or approach). Unfortunately, our study can neither unveil if what we observe are adaptive traits, nor can it say what trade-offs shaped them.

It could also be the case that sickness detection is entirely learnt and what we are observing is the work of cultural evolution. Although the latter alternative is possible, there are, as the reviewer correctly notes, clear differences between the groups that could be directly related to cultural trade-offs. In other words, even if there was a biological infrastructure to enable these traits, they are most certainly shaped by cultural practices. However, some of these differences could also be the result of more general learning mechanisms that are not related to culture-specific differences. For example, the different groups had different levels of experience with these type of faces (i.e., white Caucasian). It could be the case that with enough exposure the groups might converge in their sickness detection sensitivity and criterion. This is something that future cross-cultural studies could study by using training and exposure paradigms.

In the discussion we now write (Pg 18, ln 6-18):

“It should be noted that there was large individual variation in both detection rates and response bias within as well as between groups. Interestingly, Kurves and Wolf (2018)

also showed higher than expected variation in detection performance for facial sickness with substantially high numbers of expert-performers. They also showed large individual variation in response bias for facial sickness detection. Moreover, when they simulated potential social-learning strategies, they found individuals using a “follow-the-best-member” rule would increase both sensitivity and specificity with increasing group size, but an individual using a “follow-the-majority” rule would only increase specificity. Unfortunately, we do not know the specific learning strategies participants in our study used. Nevertheless, we believe the mechanisms shaping detection and response bias are multidimensional, including both micro, e.g., inter-personal values (Tybur, Lieberman, et al., 2020), as well as a macro-level factors, e.g., imitation (Heyes, 2012), culture-specific beliefs (Burenhult & Majid, 2011; Kruspe & Burenhult, 2019; Wnuk, 2016).“

4. Researchers like Robert Provine or Joshua Tybur have used modified faces manipulating the coloration of the sclera or using disfigured faces. In a recently published article, Tybur et al. (2020) found that most of the judgments (“inferences”) were based on low level features (e.g., coloration, skin texture). In sum, the idea of an ability to detect sickness using facial cues is misguided. We make inferences (e.g., Tybur and colleagues refer to “perceived infectability”) and we might find more or less agreement (consensus) across and within societies. For example, in Study 2, the Seri and the Maniq provide a clear example of high variability in the inferences being made, whereas the Jahai showed less variability. The Authors mentioned in page 16, “both the Maniq and Jahai attribute illness to external factors such as punishment from supernatural forces, unpleasant smells...” These reflections are very appropriate. It is a pity that the Authors did not conduct debriefing interviews to enquire about which facial cues were used by the participants to classify and discriminate sick faces. In sum, the participants made inferences and there were different levels of agreement on perceived sickness using facial cues.

Response 4:

The use of low-level features is an interesting point that we also see in our data. First, we agree that it would have been preferable to conduct debriefing interviews in all groups,

unfortunately this was not possible for logistical reasons on the ground in all communities. But we were able to do this in some communities. For example, the debriefing with the Seri and Mexican was informative in precisely the way the reviewer suggests. Specifically, both remarked that droopy eyelids were the most salient cue of sickness. This is interesting as it is what earlier studies with Swedes also found (Axelsson et al. 2018), and is also in line with the low-level features presented by Tybur et al. (2020). Thus, although we only have limited data at the moment, there is suggestive evidence that underlying "inferences" also shared across cultures, although future work would be required to definitively show this. In the discussion we now write (Pg 16, ln 6-16):

"Although this study does not address what these cues are or whether sickness cues in White-presenting faces may be more discernible than in faces from other communities, recent data from Swedish participants studying the same faces, found that low-level features like pale skin and droopy eyelids were the most reliable estimates of sickness (Axelsson et al., 2018). Interestingly, post-experimental debriefing with Seri and Mexican participants in the current study suggests droopy eyelids were particularly informative for them as well, indicating that partially similar cues were used. These results are also in line with Tybur et al. (2020) who found that most inferences people make about disfigured faces are based on low-level features like coloration. Future studies could manipulate various low-level features and conduct in-depth debriefing to better determine what facial features are particularly diagnostic across communities."

5. The last comment is related to the discrimination and classification tasks. Did the same participants complete Studies 1 and 2 using the same set of faces? If so, what was the rationale behind starting with a discrimination task first or using the same set of faces. Ideally, a classification task should come first to classify the best models (sick faces) that

could be validated in a second stage with a signal detection strategy. If different participants were randomly assigned to only of the studies, please clarify it in the manuscript.

Response 5:

We apologies for the confusion. The participants saw the same faces in both the identification and discrimination task. However, they did not start with the discrimination task (2AFC) but with the identification task (yes-no task). The reason we started with the identification task first was so the participants would not be biased the first time they saw a face. In the identification task the faces were presented one at a time so the participant could not compare directly to another face but had to decide based solely on the information given in each photo.

The reason to include the second experiment with 2AFC (discrimination task)—where the faces were presented side-by-side—was to determine whether if people failed the more demanding identification task, whether they might still be able to discriminate cues of sickness if they could compare two images of the same person side-by-side.

We found every group could identify a sick face (yes-no task) and also discriminate between them. We have clarified this in the manuscript. In the method section we now write (Pg, 10, Ln 2-12 ln):

“We developed two computer-based rating tasks (Figure 1C supplemental text S2) using two photographs each (LPS and placebo) of the 13 participants. In the first task, which was a detection task, participants had to decide whether the person in the photograph was sick or healthy following a yes-no procedure. Each photograph was shown one-by-one and participants were asked in their native language: “Is this person healthy or sick?” with the response options “sick” and “healthy”. This first task is demanding; should participants fail at this they might nevertheless be able to discriminate between a sick versus healthy face. To test this, in the second task, a discrimination task, the same participants were again shown the facial photos that were used in the first task, but this time with the LPS and placebo

photographs of the same individual side-by-side. Participants were asked to indicate which face looked sicker following a two-alternative forced-choice (2AFC) paradigm.”